# Use of Sertraline in Hemodialysis Patients

**DOI:** 10.3390/medicina57090949

**Published:** 2021-09-09

**Authors:** Alicja Kubanek, Przemysław Paul, Mateusz Przybylak, Katarzyna Kanclerz, Jakub Jan Rojek, Marcin Renke, Leszek Bidzan, Jakub Grabowski

**Affiliations:** 1Department of Occupational, Metabolic and Internal Diseases, Medical University of Gdańsk, 81-519 Gdynia, Poland; ppaul@gumed.edu.pl (P.P.); taszka@gumed.edu.pl (K.K.); marcin.renke@gumed.edu.pl (M.R.); 2Department of Developmental Psychiatry, Psychotic and Geriatric Disorders, Medical University of Gdansk, 80-282 Gdańsk, Poland; mateusz_przybylak@o2.pl (M.P.); leszek.bidzan@gumed.edu.pl (L.B.); jakub.grabowski@gumed.edu.pl (J.G.); 3Adult Psychiatry Scientific Circle, Department of Developmental Psychiatry, Psychotic and Geriatric Disorders, Medical University of Gdansk, 80-282 Gdańsk, Poland; jakubjanrojek@gumed.edu.pl

**Keywords:** depression, hemodialysis, sertraline

## Abstract

Depression and anxiety are the most common psychiatric disorders in end-stage renal disease (ESRD) patients treated with hemodialysis (HD) and may correlate with lower quality of life and increased mortality. Depression treatment in HD patients is still a challenge both for nephrologists and psychiatrists. The possible treatment of depressive disorders can be pharmacological and non-pharmacological. In our article, we focus on the use of sertraline, the medication which seems to be relatively safe and efficient in the abovementioned population, taking under consideration several limitations regarding the use of other selective serotonin reuptake inhibitors (SSRIs). In our paper, we discuss different aspects of sertraline use, taking into consideration possible benefits and side effects of drug administration like impact on QTc (corrected QT interval) prolongation, intradialytic hypotension (IDH), chronic kidney disease-associated pruritus (CKD-aP), bleeding, sexual functions, inflammation, or fracture risk. Before administering the medication, one should consider benefits and possible side effects, which are particularly significant in the treatment of ESRD patients; this could help to optimize clinical outcomes. Sertraline seems to be safe in the HD population when provided in proper doses. However, we still need more studies in this field since the ones performed so far were usually based on small samples and lacked placebo control.

## 1. Introduction

Depression and anxiety are the most common psychiatric disorders in end-stage renal disease (ESRD) patients treated with hemodialysis (HD). Their prevalence is increasing in recent years [1], reaching the rate of 20–40% for depression [2,3] and 20–52% for anxiety [4], depending on the methodological approach. According to the studies, these psychiatric disorders may correlate with lower quality of life, increased hospitalization rate, suicidal behavior, hemodialysis nonadherence, and increased mortality [5]. Despite the number of studies on this issue, major depressive disorder (MDD) in ESRD patients is still underdiagnosed, and its treatment is not optimal [6] [Table 1]. Anxiety disorders seem to be overlooked even more often [7], which is probably associated with a significant overlapping of symptoms with depression [8]. As initiation of dialysis is a large change in everyday functioning with a potential to induce or worsen psychiatric symptoms, screening for depression [9] and anxiety is suggested at the beginning of renal replacement treatment.

The possible treatment of depressive and anxiety disorders can be pharmacological and non-pharmacological. The first choice drugs for both of the above are selective serotonin reuptake inhibitors (SSRIs), including citalopram, escitalopram, fluoxetine, fluvoxamine, paroxetine, and sertraline Table 2 [10,11]. While their administration is widely considered to be safe and effective in ESRD patients, further studies are still needed in this field [12]. Non-pharmacological interventions may include psychotherapy, mindfulness meditation, and frequent HD [13].

In our article, we focus on the use of sertraline due to several limitations regarding the use of other SSRIs in ESRD patients. Fluoxetine and fluvoxamine are considered to be less efficacious [14]. Moreover, fluvoxamine has a significant impact on liver metabolism of other drugs. Administering citalopram and escitalopram bears the risk of increasing QTc interval in doses larger than 20 mg and 10 mg, respectively. These doses have been set as maximum for elderly patients [15,16], limiting therapeutical options with these drugs in many ESRD patients. While there have recently been voices raising doubts about decisions limiting the use of citalopram and escitalopram [17], these restrictions still stand. In view of the above, sertraline may seem like a relatively safe and efficient medication. However, before administering, one should consider several side effects particularly significant in the treatment of ESRD patients. These adverse effects, along with specific benefits, are discussed below in Table 3 and and Figure 1.

## 2. Sertraline—General Properties of the Medication

Studies evaluating sertraline’s safety and efficacy in ESRD patients performed so far usually had small samples and lacked placebo control [39]; therefore, in further parts of this article, references are also made to data from the general population. The European Renal Best Practice (ERBP) guidelines suggest that trials with SSRIs in patients who meet criteria for moderate Major Depressive Disorder (MDD) should last for 8–12 weeks, after which the effect of the treatment should be evaluated [43]. This is rather inconsistent with the psychiatric clinical practice when the first evaluation for depressive and anxiety patients is usually done after 4 weeks with considerations of temporizing, increasing the dose, switching to another antidepressant, or combining several drugs in patients with minimal response or safety issues [44,45].

Before starting the treatment with sertraline, no specific medical tests are required [46]. To minimize side effects, psychiatrists usually start with a subtherapeutic dose of 25 mg for one week with an increase to 50 mg after this period. These means of precaution should be especially considered in patients with coexisting anxiety, in advanced age, and those more susceptible to side effects. While doses up to 300 mg have been used, the maximum dose is considered to be 200 mg. ERBP guidelines do not suggest any need to change the dosing in stages 3–5 of chronic kidney disease (CKD), but some studies suggest that smaller doses of sertraline may be required in ESRD patients, yet the post-hemodialysis supplementation is suggested to be unnecessary [47]. The effect of the treatment can be usually seen after 4–6 weeks. The drug should be taken daily. While most psychiatrists and patients prefer it to be taken in the morning, some patients may experience sedation, and an evening dosage would be more appropriate. The general rule is that sertraline should be taken every day in the same regimen (time of day, dose splitting, with or without food). Due to a fairly long half-life time (66 h in the general population when including its less active metabolite, norsetraline) [48], missing a single dose should not result in any withdrawal side effects. This is also the reason why this drug is relatively easy to discontinue. Sertraline is a moderate inhibitor of CYP2B6 and CYP2D6 with a dose-dependent effect of increasing other drugs metabolized through these pathways [49]. Effect on other CYP450 subunits is not clinically significant, and its interactions are less potent than these of fluoxetine, fluvoxamine, or paroxetine.

Sertraline is considered to be safe to use. The adverse effects that seem to be more common while using sertraline compared to other SSRIs are diarrhea and activation [50]. In some cases, drowsiness, headache, or sexual dysfunctions may appear [50]. One should also keep in mind the antiplatelet effect of sertraline, which, while in some cases beneficial [51], may be an issue in patients with greater bleeding risk [52]. Modification of antidepressant treatment needs to be considered in some patients [53]. Regularly observed practice is also an adjustment of antiplatelet therapy dose (for instance, by reducing acetylsalicylic acid dose from 150 mg to 75 mg) after adding sertraline. This course of action, however, has no support in current literature.

Adverse effects during sertraline administration are usually mild and can be alleviated by changing the treatment regimen (lowering the dose, changing the time of administration, splitting the dose into BID (two times a day) or TID (three times a day), adding low doses of trazodone for sexual dysfunctions or insomnia, etc.) [54,55].

The general rule during antidepressant treatment is that we should struggle to achieve full remission and continue administration of the drug for the maintenance period of at least 6 months before any attempts of down-titrating or discontinuing the medication [56].

## 3. Impact on Cardiovascular System

The relation between chronic kidney disease (CKD) and cardiovascular disease is complicated and bidirectional. On the one hand presence of CKD is the risk factor for developing cardiovascular disease (CVD). In the meta-analysis that enrolled more than 100.000 participants, reduction in both eGFR and albuminuria was associated with all-cause mortality and cardiovascular mortality independently of each other and traditional cardiovascular risk factors [57]. On the other hand, cardiovascular disease remains the major cause of morbidity and mortality in patients with end-stage renal disease (ESRD). In general, the prevalence of cardiovascular diseases among ESRD patients who received HD was 70.6%, being also high (>50%) among younger patients between 22 and 44 years of age, according to US Renal Data System 2018 [58]. Besides the existence of traditional coronary artery disease (CAD) risk factors, such as diabetes and hypertension, patients with CKD are also exposed to other non-traditional, uremia-related cardiovascular disease risk factors, including inflammation, oxidative stress, and abnormal calcium—phosphorus metabolism. HD patients can be asymptomatic or manifest atypical symptoms of a chronic coronary syndrome (CCS), mimicking symptoms of fluid overload or intradialytic hypotension (IDH). Furthermore, patients undergoing dialysis due to CKD were frequently excluded from coronary artery evidence-generating clinical trials [59], which can also influence outcomes in this population. Taking into account the abovementioned limitations, the proper diagnosis and therapy of ESRD patients with CVD can be limited both by the safety issues and the insufficient data. Depression is associated with increased disease burden and with a higher risk of all-cause mortality and cardiovascular mortality worldwide [60,61]. The meta-analysis that included a total of 293 studies with 1,813,733 participants from 35 countries confirmed the presence of association between depression and excess mortality [60].

The effect of antidepressant treatment on the cardiovascular system may vary depending on the agent used and type of disease. SSRIs can cause side effects such as QTc prolongation [62] or bleeding [63], but they can also have some desired effects like reduced platelet reactivity [23,24], Table 3. SSRIs are relatively safe and are the first-line therapy for patients with heart failure [64], although the efficacy of antidepressant therapy on the outcomes of patients with HF is controversial [65]. SSRIs are also the first-line treatment for pharmacological management of MDD in CAD patients since they do not have the cardiotoxic adverse effects of tricyclic antidepressants [66]. Due to its low risk of drug-drug interactions, adverse effect profile, and potential for beneficial antiplatelet activity, sertraline could be considered the first-line choice antidepressant for patients with CAD [67].

Patients dependent on maintenance hemodialysis are at high risk of arrhythmia due to the comorbidities, high level of polypharmacy, and electrocyte disturbances. SSRIs can cause prolongation of QTc interval, with citalopram and escitalopram having the greatest impact [62,68] with the effect increasing with the dose [69]. Other SSRIs, including sertraline, are associated with QTc prolongation but seem to be safe when provided in the recommended doses (Table 3). In the retrospective cohort study, Assimon et al. compared two groups of SSRIs: those with higher potential for prolonging the QT interval and those with lower potential (citalopram and escitalopram versus fluoxetine, fluvoxamine, paroxetine, and sertraline, respectively). The authors observed that the initiation of a higher versus lower QT-prolonging–potential SSRI was associated with a higher risk of sudden cardiac death in a cohort of patients on hemodialysis. Elderly patients, females, patients with conduction disorders, and those treated with other non-SSRI QTc-prolonging medications were the most endangered ones [20]. The results emphasize the importance of individualization of the therapy. More studies are still needed to clearly understand the safety profile of SSRIs in the dialysis population.

The hemostatic disorders in the ESRD population are complex, and patients can experience two opposite complications: bleeding diathesis and thrombotic tendency. One of the mechanisms causing the bleeding diathesis is platelet dysfunction and impaired platelet–vessel wall interaction. Dialysis reduces the risk of hemorrhage by improving platelet abnormalities induced by toxins, but it can also contribute to bleeding by the platelet activation during the hemodialysis sessions [70]. Antidepressants with potent serotonin reuptake inhibitor (SRI) activity increase the risk of bleeding through different mechanisms, and the upper gastrointestinal (GI) tract is the commonest site of SRI-related abnormal bleeding [63]. The meta-analysis by Jiang et al. that included a total of 22 studies (6 cohort and 16 case-control studies) involving more than 1,073,000 individuals revealed that SSRI use was associated with an almost 2-fold increase in the risk of developing upper GI bleeding, especially among patients at high risk for GI bleeding (concurrent use of nonsteroidal anti-inflammatory or antiplatelet drugs) [21]. CKD is associated with increased risk of bleeding in both operative and non-operative patients [25,71]. The large population-based study by Iwagami et al. demonstrated that while the relative risk of GI bleeding associated with SSRI exposure was constant, the excess risk of GI bleeding associated with SSRIs markedly increased among patients with decreased kidney function [22], Table 3.

On the other hand, one of the potential mechanisms contributing to excessive CV (Cardiovascular) risk is increased platelet reactivity [72] and excessive stickiness of endothelial cells to platelets [70] in both CKD [72] and MDD populations [73]. Some data suggest that treatment of MDD with SSRI may reduce platelet aggregation and activation markers [23,24,25]. In the recently published, randomized, double-blind trial, Jain et al. analyzed the whole blood platelet aggregation (WBPA), plasma levels of E-selectin and P-selectin on treatment with sertraline vs. placebo. The study group consisted of 175 participants with CKD, and the observation period lasted for 12 weeks. The results revealed the increased adenosine diphosphate (ADP)-induced platelet aggregability in CKD patients compared to controls with normal kidney function, regardless of the presence of comorbid MDD. Treatment with sertraline did also not affect platelet function. These findings suggest that increased platelet activation may not be a major pathological mechanism by which depression leads to worse cardiovascular outcomes in patients with CKD. Future studies should include positive MDD controls without CKD to confirm these findings [74].

## 4. Intradialytic Hypotension

Intradialytic hypotension (IDH) is a common complication of hemodialysis (HD), which worsens the patient’s outcome. It is reported to occur in about 20% of HD sessions [75,76]. The definition of IDH is not standardized and differs between various sources. According to the National Kidney Foundation Disease Outcomes and Quality Initiative (KDOQI), IDH is a drop in systolic blood pressure (SBP) or mean arterial pressure (MAP) associated with symptoms such as vertigo, weakness, abdominal or chest pain, muscle cramps, nausea, and paleness [77]. On the other hand, Flythe et al. analyzed various definitions of IDH. They showed that an absolute nadir SBP < 90 mmHg or <100 mmHg (depending on pre-HD SBP), regardless of the symptoms, was most potently associated with mortality [78]. Thus, nadir-SBP based definitions are gaining popularity in the literature [79].

IDH is proved to be an independent risk factor for mortality in HD patients [80]. HD- induced myocardial stunning may contribute to a higher prevalence of cardiovascular events in this population [81]. IDH may also have negative effects on other organs, such the brain and the GI tract [79,82].

Pathophysiological mechanisms involved in IDH development are complex. In general, hypotension during HD occurs when compensatory systems controlled by sympathetic stimulation (such as cardiac output, plasma refill, and peripheral vascular resistance) are unable to equal decreased effective plasma volume due to hemofiltration and decreased extracellular osmolality associated with sodium removal [79,82].

One of several pharmacological agents, which have been tested in the prevention of IDH, is sertraline. This SSRI has improved symptoms in refractory neurocardiogenic syncope [83]. Both of these conditions seem to have a similar pathogenic mechanism, which is the paradoxical sympathetic vasoconstrictors withdrawal owing to rapid serotonin release in the central nervous system. It is postulated that sertraline may induce down-regulation of postsynaptic serotonin receptors through intensification of serotonergic transmission and thereby moderate the potential response to the sudden increase in cerebral serotonin levels in response to hypovolemic stress [83].

Molin et al. conducted a study in which they examined the efficacy of sertraline in the prevention of IDH. The analyzed group consisted of 16 hemodialysis participants who were observed for 12 weeks. They described dialysis-induced hypotension as a drop of SBP ≥ 30 mmHg during the procedure or pre-hemodialysis SBP ≤ 100 mmHg accompanied by symptoms: weakness, cramps, dizziness, headache, blurred vision, nausea, vomiting, and malaise; any SBP ≤ 90 mmHg and/or DBP ≤ 40 mmHg regardless the symptoms; or any patient with symptoms listed above who required nursing intervention. Participants enrolled on the study had IDH in at least 50% of dialysis sessions during a 3-month period preceding SSRI use. Comparison of IDH prevalence between sertraline and placebo groups revealed no statistical difference. Nevertheless, the authors showed that the risk of reporting intradialytic symptoms was 42% higher in the placebo group [26]. However, as Georgianos and Agarwal commented in their paper, the study of Molin and colleagues may suffer from some methodological limitations, such as a small number of participants, single-blind, and no cross-over design. Furthermore, they conclude that sertraline, as a mood-enhancing agent, may be beneficial to reduce the number of uncomfortable intradialytic symptoms and nursing interventions [84].

Another clinical trial assessing sertraline in the prevention of IDH was designed by Razeghi and colleagues as a randomized, double-blind, cross-over prospective study. Inclusion criteria were similar to those mentioned in the previous paragraph. The study group consisted of 12 participants who were observed for 12 weeks total (4 weeks of sertraline, 4 weeks of washout period, and 4 weeks of placebo). They showed that sertraline therapy significantly reduced the risk of hypotension episodes by 43% and increased postdialysis systolic and diastolic BP by 8.7 mmHg and 6.0 mmHg, respectively, compared with the placebo group. Moreover, it decreased the total number of IDH interventions (although not significantly). The researchers pointed out that the positive effect of sertraline in the prevention of IDH was only seen in patients without diabetes mellitus (DM) [27].

Yalcin et al. designed two studies testing sertraline in hemodialysis patients with IDH. The first one was performed on nine patients who had been suffering from IDH (defined as at least 30 mmHg drop in SBP or SBP < 100 mmHg accompanied with symptoms) for at least 3 months. No placebo control group was included. They compared the data of the 4-week pre-sertraline period with a 4-week sertraline period. The nadir SBP and nadir DBP during hemodialysis were significantly higher, while the number of therapeutic interventions declined significantly during the sertraline part [28]. This pilot study was followed by a placebo-controlled, prospective study on 30 patients. The exclusion criteria included DM, amyloidosis, and structural heart disease in order to avoid autonomic dysfunction. The results were analogous to the pilot study [29], Table 3.

On the contrary, Brewster and colleagues did not prove any positive effect of sertraline in the prevention of IDH. A total of 18 patients enrolled on their prospective, cross-over study had IDH defined as at least three episodes of decreased SBP associated with symptoms in 50% of HD sessions over a 1-month period. Nadir SBP and DBP during HD, as well as MAP, did not differ significantly between the control phase and the sertraline phase of the study. Among possible explanations of their results, the authors enumerate population with resistant IDH enrolled on their study, the insufficient dose of sertraline, comorbidities, especially DM, which could cause diabetes-associated autonomic neuropathy, and therapy with midodrine (selective, α-1 adrenergic agonist) [30].

In conclusion, IDH is a common and dangerous complication of HD procedure. It is of great importance to find appropriate measures to face this problem. The abovementioned publications present inconsistent results, and in our interpretation, they suffer from some methodological limitations, especially low sample sizes. Despite promising results, further studies should be designed to examine the efficacy of sertraline in the prevention of IDH.

## 5. Uremic Pruritus

Uremic pruritus is a frequent and burdensome symptom affecting patients with advanced CKD. Chronic kidney disease-associated pruritus (CKD-aP) is defined as itching directly related to kidney disease, without another comorbid condition to explain itching [85]. Its severity, location, and time of onset may vary significantly, and it can occur without accompanying skin changes [86]. Due to the lack of an unambiguous definition and precise measurement tools, its frequency is difficult to determine. In the international prospective cohort Dialysis Outcomes and Practice Patterns Study (DOPPS), 13–50% of dialysis patients reported suffering from moderate to extreme pruritus differing between countries [87,88]. Furthermore, data from mentioned above DOPPS study revealed that patients with moderate to severe pruritus had significantly higher mortality rates. Uremic itch can contribute to mood disturbances and can be associated with a decreased quality of life (QoL) [87]. The study by Mathur et al. showed that the intensity of the pruritus was strongly associated with diminished health related-QoL in multiple domains, including mood, sleep, and social relations [89].

The etiology of itch in the HD population is complex. Certain patients’ characteristics and dialysis parameters are correlated with the prevalence of uremic pruritus. This can include dialysis adequacy, low-flux dialyzers, hepatitis C positivity, higher CRP, calcium/phosphorus levels, older age, and underlying depression [90]. The pathogenesis of uremic pruritus is incompletely understood, though there are theories considering toxin deposition, peripheral neuropathy, immune system dysregulation, and opioid imbalance [85].

Pruritus patients suffer more frequently from depression, and the depressive state can amplify itch perception [91]. Oral antidepressants are thought to have an antipruritic effect due to their influence on serotonin and histamine levels, and they can be effective in cases not responding to other therapies and were recommended in the European Guideline for Chronic Pruritus [92].

In the systematic review by Kouwenhoven et al., a total of 35 studies evaluating the oral use of SSRIs, tricyclic antidepressants (TCAs), and/or atypical antidepressants in chronic pruritus published between 1980 and 2016 were included [93]. The majority of the analyzed studies showed the improvement of pruritus after initiation of oral antidepressant therapy. Both paroxetine and mirtazapine can be effective options in pruritus due to malignancies [94,95]. Furthermore, the cross-over randomized clinical trial by Gholyaf et al. showed that mirtazapine could be effective for UP patients who are on maintenance hemodialysis [96].

In cases of pruritus caused by cholestasis or CKD, sertraline could be an effective treatment option [31,32,97]. Two studies evaluating the use of sertraline in patients with CKD suffering from pruritus revealed a significant reduction in itching [31,32], Table 3. In an open-label trial by Shakiba et al., 19 patients undergoing HD suffering from UP were treated with sertraline in the medium dose of 50 mg daily for four months with resulted in the reduction of itch in the majority of cases [31]. In the second mentioned above retrospective cohort study conducted in the renal palliative care clinic, the treatment of sertraline in the medium effective dose of 35 mg daily was introduced in 17 patients. The study showed the effectiveness of low-dose sertraline in patients with antihistamine-refractory uremic pruritus measured using the VAS (Visual Analogue Scale) score [32]. In the double-blinded clinical study, which gathered 50 patients treated with HD and suffered from pruritus, half received sertraline in the dose of 50 mg twice a day, and half received placebo, patients were observed for the period of 6 months. The results of a decrease in itch intensity were more significant in the sertraline group compared to control. The authors emphasize the role of inflammation in uremic itching and suggest that sertraline can be an effective drug in reducing itch also due to its effect on reducing inflammatory cytokines [33].

Although the problem of CKD-aP is common, more evidence on how to optimize the treatment is still needed. The non-pharmacological approach is based on dialysis modification (using the biocompatible membranes, normalizing the substances that may aggravate itching) and phototherapy. Treatment trials of pruritus included different medications, such as antihistamine agents, cyclosporin, pregabalin, and many others. According to the previous research, gabapentin seems to be efficacious and safe in improving uremic pruritus among dialysis patients, and therefore it can be considered as a drug of the first choice [98]. The new peripherally restricted and selective agonist of kappa opioid receptors, difelikefalin, is under research and was shown to significantly reduce the itch intensity and improve quality of life compared to placebo [99].

Even though sertraline is not the first-choice therapy of the uremic pruritus, it can be considered in cases resistant to other medications, especially in the group of dialysis patients requiring antidepressant treatment at the same time. However, more studies on the effectiveness of SSRIs in UP are needed since the available evidence is insufficient.

## 6. Cytokines

Advanced CKD is often associated with chronic inflammation, which can significantly increase patient mortality [100,101,102]. Dialysis patients die 10–20 times more often from cardiovascular causes compared with the general population [103]. The probable cause of an increased CVD risk in this group is a dysfunction of vascular endothelium, which promotes the atherosclerotic process [104]. In the group of dialysis patients, an increased concentration of C-reactive protein (CRP) and pro-inflammatory cytokines such as interleukin-1β (IL-1β), interleukin-6 (IL-6), tumor necrosis factor-α (TNF-α), chemotactic factor for monocytes-1 (MCP-1), and hepatocyte growth factor (HGF) are observed, which is one of the suggested causes of damage to the vascular endothelium [105,106,107,108].

One of the proposed etiologies of depression is the inflammatory hypothesis—the presence of an inflammatory process and activation of the immune system in the peripheral and central nervous systems [109]. According to it, pro-inflammatory cytokines have the ability to modulate the synthesis, release, and turnover of monoamines, which are crucial in the regulation of mood [109]. Consequently, prolonged and severe depressive symptoms may be accompanied by an excessive concentration of pro-inflammatory cytokines such as IL-1β, IL-6, and TNF-α [110]. The study by Yamasaki et al., published in March 2020, also suggests a significant increase in the concentration of soluble interleukin-6 receptor (sIL-6R) in people diagnosed with major depressive episode [111].

Both renal failure in the course of CKD, which requires renal replacement therapy (RRT), and depression are disease processes with an increased inflammatory response [100,101,102,109,112]. However, there are few publications at the moment that assess the correlation between patient’s clinical condition, degree of renal failure, concentration of pro-inflammatory cytokines in the blood serum, and the severity of depressive symptoms. Finding a relationship between them would allow determining the prognosis of the progression of the inflammatory process on the functioning of this group of patients and could indirectly answer whether antidepressant treatment resulting in the reduction of biochemical markers of inflammation could reduce the process of damage to the vascular endothelium responsible for the main cause of mortality in dialysis patients [104].

Sertraline, as well as other drugs from the group of SSRIs, is one of the most commonly used antidepressants with proven therapeutic efficacy [113,114]. At the same time, it seems to be a safe drug in people with a high risk of cardiovascular complications [115,116]. In addition, various scientific studies have shown that the use of sertraline reduces the concentration of inflammatory markers, such as CRP or pro-inflammatory cytokines—IL-1β, IL-6, and interleukin-12 (IL-12), while increasing the level of anti-inflammatory cytokines, such as interleukin- 4 (IL-4) and transforming growth factor-β1 (TGF-β1) [34,35,36], Table 3. Therefore, it seems that sertraline may be a substance that, apart from its antidepressant effect, also reduces the risk of cardiovascular complications caused by chronic inflammation, which in turn may reduce mortality in patients treated with RRT [117]. However, there is still a lack of a sufficient number of publications that could prove the efficacy and safety of sertraline in CKD dialysis patients with depressive symptoms [118,119] or estimate the role of other antidepressants in this group of patients.

## 7. Sexual Dysfunction

The data on sexual dysfunctions (SD) in CKD patients are ambiguous. Some scientists state that the problem is underestimated [120,121], while others depict it as a marginal phenomenon [122]. When evaluating the severity of this issue, one needs to take into consideration that patients more often disclose sexual dysfunction symptoms while being directly asked rather than spontaneously report them [123].

In a meta-analysis by Navaneethan et al., sexual life of females and males with CKD was explored. Mainly studies using validated tools such as the International Index of Erectile Function or Female Sexual Function Index (FSFI) were taken into consideration, with a total of over 8343 patients evaluated. Erectile dysfunction was the most prevalent disorder observed in men with CKD (70%). Data from studies on women were not sufficient to perform a meta-analysis, but in general, females with CKD had a significantly lower overall FSFI score than women from the general population reporting disturbances in desire, arousal, lubrication, orgasm, and satisfaction as well as coital pain. The study highlighted that increasing age, diabetes mellitus (DM), and depression also correlate with sexual dysfunction and may generally aggravate problems with sexual life quality in patients with CKD. A high prevalence of sexual disorders was observed, especially in subjects receiving dialysis [124].

Possible pathophysiological explanations of these disorders involve toxicity of high urea level, psychological factors and endocrine disorders, such as abnormalities in the hypothalamic-pituitary axis (more specific in females), and hypogonadism (a major role in males) [125,126].

Sexual dysfunction is one of the frequent problems affecting people with CKD [127,128], but it may also be a symptom of MDD alone [129,130,131]. Furthermore, sexual disorders and depression during CKD can also develop into a mechanism of the vicious circle. Antidepressants may improve sexual functioning by alleviating MDD symptoms. On the other hand, decreased libido, delayed orgasm, anorgasmia both in men and women, erectile dysfunctions, decreased lubrication, and lower general sexual satisfaction can also be side effects of such therapy [37,130,132], especially with SSRIs.

Sertraline is considered to have a significant negative impact on sexual functioning in the general population [37,38]. Data on its potential in causing SD in the group of CKD patients are scarce, with very low quality of evidence showing a surprisingly low prevalence of SD both in AD treatment and in the placebo groups [39], Table 3. In case of medication-related sexual dysfunctions in patients not affected by CKD, we may consider changing sertraline (or other administered SSRI) to other medications with a lesser risk of causing SD (vortioxetine, bupropion) or by adding a small dose of an antidepressant with a different mechanism of action (trazodone, agomelatine) enabling us to lower sertraline dose and decrease this dose-dependent adverse effect [123,132,133,134,135,136]. We have found no studies regarding vortioxetine in CKD. Agomelatine and trazodone do not require dose adjustment in ESRD patients, while the maximum dose of bupropion should not exceed 150 mg [43]. In patients with CKD, several other safety issues may restrict the use of medications [137], and introducing any of the above strategies requires an individual approach. Further studies are needed to evaluate the safety and efficacy of other interventions to cope with sexual dysfunctions in CKD patients.

## 8. Fracture Risk

The use of SSRIs is associated with hip fracture risk in the general population [40,41], but the impact of antidepressants on this complication in the population of hemodialysis (HD) patients is complex and not fully understood. Patients with end-stage renal disease (ESRD) treated with HD have four times higher rates of hip fracture compared with the general population [138,139], which can be related to the frequent occurrence of acidosis, mineral and bone disorders, dysautonomia, cachexia, inflammation, and other comorbid conditions. Furthermore, patients with ESRD treatment with HD have an increased risk of falls and fall-related injuries. The rate of falls was found to be 1.18–1.60 falls/year, which is significantly higher than in seniors not treated by HD [140,141].

Depressive disorders are also an independent risk factor for falls and fall-related injuries among the elderly population [142,143,144,145,146]. As mood disorders are more likely to be reported in patients with CKD than in the general population or in adults with other chronic conditions, depression may also be an important fall risk factor in this group.

Kistler et al. analyzed 16,573 individuals with CKD through The Behavioral Risk Factor Surveillance System (BRFSS) data to find an association between falls, depression, and CKD [147]. It was revealed that despite adjusting for multiple confounders, history of depression was correlated with falls and fall-related injuries. In previous studies, Kistler and colleagues already showed that patients with CKD are more likely to fall and suffer fall-related injuries compared with adults without CKD [148,149]. The factors that may be responsible for the elevated fall risk in patients with CKD are protein and energy wasting, muscle weakness [150,151], cognitive impairment, polypharmacy, hemodynamic instability, and high rates of comorbidities such as cardiovascular diseases and diabetes [152]. Kidney impairment contributes to changes in bone metabolism and leads to bone alterations and extraskeletal calcification, which may increase the risk for injury when suffering a fall [153,154]. Research available shows an effect of the entire group of SSRIs drugs without detailing the effect of sertraline on hip fractures.

Vangala et al. examined the relationship between SSRIs consumption and hip fracture occurrence among patients with ESRD treated with HD, a unique high-risk subpopulation, within which the impact of SSRIs on hip fracture risk remains unexplored [42]. They identified 4912 cases by use of the US Renal Data System (USRDS), the national registry of persons with ESKD treated with RRT (renal repacement therapy), to conduct a case-control study nested in the recorded person-time between 1 January 2006 and 30 September 2015 compared to 49,120 controls. SSRI use, including sertraline, was associated with increased hip fracture risk (adjusted OR (odds ratio), 1.25; 95% CI (Confidence interval), 1.17–1.35). Risk for fracture was estimated for any, low, moderate, and high SSRI use: adjusted conditional ORs were 1.25 (95% CI, 1.17–1.35), 1.20 (95% CI, 1.08–1.32), 1.31 (95% CI, 1.18–1.43), and 1.26 (95% CI, 1.12–1.41), respectively. The association between hip fracture events and SSRI use, including sertraline, has also been seen in the examination of new short-term use (adjusted OR, 1.43; 95% CI, 1.23–1.67). The stronger association with new short-term use may suggest an acute mechanism potentially related to falls. The study revealed that patients treated by maintenance HD who suffered from a hip fracture used SSRIs more commonly than controls Table 3.

As similar associations have been repeatedly demonstrated in the general population, particular attention should be paid early for any potential side effects assigned to sertraline (hyponatremia, orthostasis, QTC prolongation, and arrhythmias) that may contribute to fall risk. European Renal Best Practice Guidelines suggest that in patients with stage 3–5 CKD, treatment effect should be reassessed after 8–12 weeks to avoid “prolongation of drugs ineffective,” and this view should also be taken into consideration for patients with ESRD treated with RRT.

The moderate increase in hip fracture risk associated with sertraline, similar to other SSRIs use, is worth considering, especially if patients and physicians do not perceive benefit from the medication after an appropriate duration of use.

There is evidence that sertraline, similar to other SSRIs, may decrease bone mass, and as a consequence, intensifies the risk of osteoporosis and osteoporosis-related fractures. This is significant since the dominant indication of SSRIs is depression, a condition also associated with low bone mass, osteoporosis, and nonpathological fractures. Considering that SSRIs, including sertraline, are the most prescribed antidepressants in the world, a large number of persons at risk of osteoporosis will be treated with SSRIs, which means people with an already increased risk of fractures are exposed to even greater risk. Furthermore, many of the risk factors for developing depression overlap with those for osteoporosis, such as physical inactivity, poor diet, and smoking [155].

## 9. Discussion

Depression and anxiety are the most common psychiatric disorders in the end-stage renal disease patients treated with HD and may correlate with lower quality of life and increased mortality [5]. Depression treatment in hemodialysis patients is still a challenge both for nephrologists and psychiatrists. In our article, we focused on the use of sertraline, the medication which seems to be relatively safe and efficient in the abovementioned population. However, we still need more studies in this field since the ones performed so far were usually based on small samples and lacked placebo control. In our paper, we discuss different aspects of sertraline use, taking into consideration possible benefits and side effects of drug administration (Table 1, Table 2 and Table 3), which could be helpful in optimizing clinical outcomes.

## 10. Conclusions

Depression and anxiety are the most common psychiatric disorders in end-stage renal disease (ESRD) patients treated with hemodialysis (HD). They may correlate with lower quality of life and increased mortality in the abovementioned population. Despite the importance of the problem, the knowledge of antidepressants usage in this population is still limited. In our paper, we reviewed previous studies considering the use of sertraline in hemodialysis patients, the advantages of using the drug, and the possible dangers of treatment. We believe that such a summary can be helpful for clinicians in their daily work with hemodialysis patients.

## Figures and Tables

**Figure 1 medicina-57-00949-f001:**
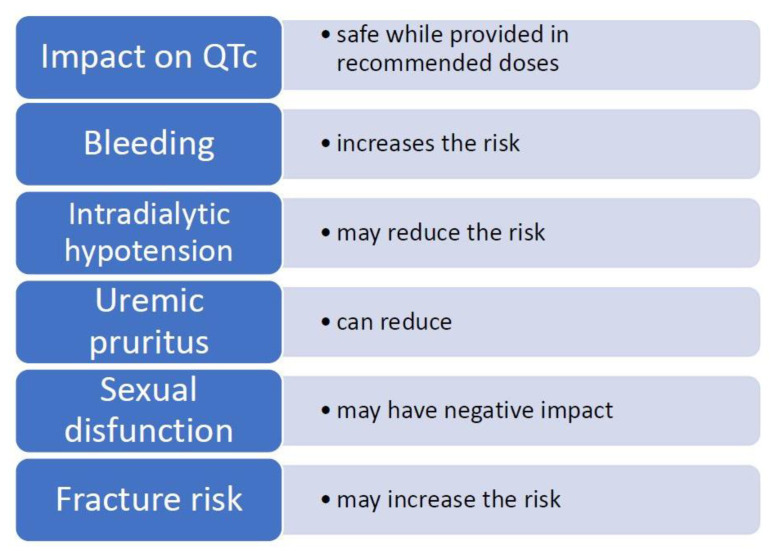
Possible effects of sertraline in the hemodialysis patient.

**Table 1 medicina-57-00949-t001:** Comparison of depression in ICD-10 and DSM-5 classifications.

Subject of Comparison	ICD-10 Classification	DSM-5 Classification
Nomenclature	Depressive Episode [18]	Major depressive disorder (MDD) [19]
Main symptoms	Depressed mood to the degree that is definitely abnormal for the individual, present for most of the day and almost every day, largely uninfluenced by circumstances Loss of interest or pleasure in activities that are normally pleasurable Decreased energy or increased fatiguability [18]	Depressed mood most of the day, nearly every day, as indicated in the subjective report or in observation made by others Markedly diminished interest in pleasure in all or almost all activities most of the day and nearly every day
Additional symptoms	Loss of confidence and self-esteem Unreasonable feelings of self-reproach or excessive and inappropriate guilt Recurrent thoughts of death or suicide, or any suicidal behaviour Complaints or evidence of diminished ability to think or concentrate, such as indecisiveness or vacillation Change in psychomotor activity with agitation or retardation (either subjective or objective) Sleep disturbance of any type Change in appetite (decrease or increase) with corresponding weight change) [18]	Significant weight loss when not dieting or weight gain, for example, more than 5 percent of body weight in a month or changes in appetite nearly every day Insomnia or hypersomnia nearly every day Psychomotor agitation or retardation nearly every day Fatigue or loss of energy nearly every day Feelings of worthlessness or excessive or inappropriate guilt Diminished ability to think or concentrate, or indecisiveness nearly every day Recurrent thoughts of death [19]
Diagnostic criteria	At least two main symptoms and additional symptoms in a total number of at least four [18]	At least one main symptom and additional symptoms in a total number of at least
Duration of symptoms	At least two weeks [18]	At least two weeks [19]
Severity of symptoms	Clinical differentiation: Mild Depressive Episode (at least two main symptoms and additional symptoms with a total number of at least four) Moderate Depressive Episode (at least two main symptoms and additional symptoms with a total number of at least six) Severe Depressive Episode (all three main symptoms and additional symptoms with a total number of at least eight) [18]	The symptoms ought to cause significant impairment in social, occupational or another important area of functioning [19]
Exclusion criteria	No hypomanic or manic symptoms sufficient to meet the criteria for hypomanic or manic episode at any time in the individual’s life The episode is not attributable to psychoactive substance use The episode is not attributable to any organic mental disorder [18]	No preceding manic or hypomanic episode The disorder is not connected with psychological effects of the substance use or another medical condition The disorder cannot be better explained by schizophrenia or any other psychotic disorder [19]

**Table 2 medicina-57-00949-t002:** Antidepressive agents with serotonin reuptake transporter (SERT) inhibition activity.

SSRI	SNRI	SARI	SNRISA	SMS	TCA
Citalopram	Desvenlafaxine	Nefazodone	Amoxapine	Vilazodone	Amitriptyline
Escitalopram	Duloxetine	Trazodone		Vortioxetine	Clomipramine
Fluoxetine	Levomilnacipram				Desipramine
Fluvoxamine	Milnacipran				Doxepin
Paroxetine	Venlafaxine				Imipramine
Sertraline					Nortriptyline
					Protriptyline
					Trimipramine

SARI—serotonin receptors antagonist with serotonin reuptake inhibition; SMS—serotonin modulator and stimulator; SNRI—serotonin–norepinephrine reuptake inhibitors; SNRISA—serotonin–norepinephrine reuptake inhibitor and serotonin receptors antagonism antidepressant with potent antipsychotic D2 receptor blockade/antagonism; TCA (tricyclic antidepressants).

**Table 3 medicina-57-00949-t003:** Effects of sertraline use in hemodialysis patients.

Impact on QTc Prolongation	Safe While Provided in Recommended Doses [20]
Bleeding	Increases the risk of bleeding [21,22]
Platelet reactivity	May reduce platelet activation [23,24,25]
Intradialytic hypotension (IDH)	Inconsistent study results, may reduce the risk of IDH [26,27,28,29,30]
Chronic kidney disease-associated pruritus (CKD-aP)	Can reduce pruritus in cases caused by CKD [31,32,33]
Cytokines	Reduces the concentration of pro-inflammatory and increases the levels of anti-inflammatory cytokines, insufficient data in HD population [34,35,36]
Sexual disfunction (SD)	Negative impact on SD in general population [37,38] and in HD population [39]
Fracture risk and osteoporosis	Is associated with hip fracture risk and may decrease bone mass in general population [40,41], may increase fracture risk in ESRD population [42]

HD (hemodialysis); ESRD (end-stage renal disease).

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
