# Peer review of "Use of Sertraline in Hemodialysis Patients"

_medicina, 2021, doi:10.3390/medicina57090949_

Round 1
Reviewer 1 Report
The principal suggestion that I have is that tables need to be analyzed and discussed deeply, without an analysis data is not useful.
In my perspective the review is very general, do not analyze the information deeply.
The article could be more interesting if the physiological effect is included, moreover because of the last studies it will be interesting if transcriptomic effects are analyzed, the effect in the expression of certain genes, enzymes that could be traduced to the physiological effects.
Author Response
Point 1: The principal suggestion that I have is that tables need to be analyzed and discussed deeply, without an analysis data is not useful.
Response 1: Thank you for this comment.
According Table 1: It summarizes individual issues raised in the text, it also consists links to articles included in the references. There is a reference to the table in the end of the text.
We added new references to the table (page 2,3,4,5,6,8,10).
According Table 2: We added the reference to the table in the text where the data from the table is mentioned (page 1)
According Table 3: We added the reference to the table in the text where the data from the table is mentioned to make it more clear (page 1)
Point 2: In my perspective the review is very general, do not analyze the information deeply.
Response 2: Thank you for this comment.
We are aware that the manuscript is general in nature, largely due to the extend of the issue. In addition, we wanted to show various aspects of the use of sertraline in hemodialysis patients in a cross-sectional way, which could be valuable for both nephrologists and psychiatrists.
Point 3: The article could be more interesting if the physiological effect is included, moreover because of the last studies it will be interesting if transcriptomic effects are analyzed, the effect in the expression of certain genes, enzymes that could be traduced to the physiological effects.
Response 3: Thank you for this comment.
The topic is very interesting, but it seems to us that it goes beyond the scope of our issue. Extending our text to include the transcriptomic effects would require additional time, exceeding the established limit. It would be interesting to analyze it in the next paper.
Reviewer 2 Report
Kubanek et al submit "Use of sertraline in hemodialysis patients.". They look at effects of sertraline in Depression and anxiety in the end-stage renal disease (ESRD) patients treated with hemodialysis (HD). They conclude that Sertraline seems to be safe in the HD population while provided in proper doses.
Page 2 , "In some cases, drowsiness, headache or sexual dysfunctions may appear. " needs a reference.
The paper is well written but needs a recapitulative figure
Minor
in abstract correct cosideration and define QTc
correct "in stages 3 to5 " page2
in table 1 correct "inconsisatnt"
Author Response
Point 1: "In some cases, drowsiness, headache or sexual dysfunctions may appear. " needs a reference.
Response 1: Thank you for this comment.
We added the reference- Hu XH, Bull SA, Hunkeler EM, et al. Incidence and duration of side effects and those rated as bothersome with selective serotonin reuptake inhibitor treatment for depression: patient report versus physician estimate. J Clin Psychiatry 2004; 65:959.
Point 2: The paper is well written but needs a recapitulative figure
Response 2: Thank you for this comment.
I added the recapitulative figure basing on the Table 1 content. I chose the most practical information that could be useful while working wit hemodialysis patient.
Point 3: Minor
1. in abstract correct cosideration and define QTc
2. correct "in stages 3 to5 " page2
3. in table 1 correct "inconsisatnt"
Response 3:
ad.1. in abstract correct cosideration and define QTc> corrected
ad.2 correct "in stages 3 to5 " page2> corrected
ad.3 in table 1 correct "inconsisatnt"> corrected

Round 2
Reviewer 1 Report
I suggest accepting the paper in the present form
Reviewer 2 Report
changes are ok